# One-Shot Federated Distillation Using Monoclass Teachers: A Study of Knowledge Fragmentation and Out-of-Distribution Supervision

**Cedric Maron**                                            *cedric.maron@segula.fr*
*Segula Technologies, France*

**Virginie Fresse**                                  *virginie.fresse@univ-st-etienne.fr*
*Hubert Curien Laboratory, Université Jean Monnet, Saint-Étienne, France*

**Mathieu Orzalesi**                                        *mathieu.orzalesi@segula.fr*
*Segula Technologies, France*

**Reviewed on OpenReview:** *https://openreview.net/forum?id=ENdm5BM7aF*

## Abstract

The performance of machine learning models critically depends on the quality and diversity of training data. However, privacy, legal, and proprietary concerns often limit direct data sharing. Many organizations possess high-quality data for specific classes and may wish to share the knowledge derived from it without revealing the data or engaging in collaborative training. While federated learning (FL) enables distributed model training, it typically assumes mutual benefit, requires repeated communication, and produces a shared global model. Another paradigm, knowledge distillation (KD), allows a student model to learn from teacher predictions. We propose a one-shot federated distillation method in which a single client learns from monoclass teacher models trained independently by multiple providers. Each provider shares its model once, and the client combines these with unlabeled data to distill a multiclass student model—aggregating knowledge from disjoint, class-specific sources. This unidirectional, asymmetric setup poses a key challenge: out-of-distribution (OOD) supervision, where monoclass teachers often mispredict unseen inputs, leading to noisy signals for the student. The main contribution of this work is a systematic study of knowledge fragmentation in one-shot federated distillation with monoclass teachers. We evaluate five configurations with varying class coverage per provider and show that increasing fragmentation intensifies OOD supervision, degrading student performance. Experiments on MNIST, FashionMNIST, and CIFAR-10 confirm that fragmentation consistently reduces student accuracy. To mitigate this, we discuss three strategies: (1) exposing teachers to diverse off-class examples, (2) penalizing overconfidence, and (3) using contrastive learning to sharpen feature boundaries.

## 1 Introduction

Modern machine learning models rely heavily on diverse, high-quality training data. Yet in real-world scenarios, relevant data is often siloed across organizations, each holding large datasets focused on specific classes or domains. Privacy, legal, and proprietary constraints frequently prohibit raw data sharing. Many organizations are open to sharing learned knowledge, either freely or commercially, particularly when they hold more data of higher quality than others. This creates an opportunity for under-resourced clients to benefit from distributed, class-specific expertise without requiring direct collaboration.

Federated learning (FL) offers a decentralized solution for training models across distributed data sources without sharing raw data. However, typical FL protocols require multiple rounds of communication, syn-

chronized participation, and shared training objectives assumptions that break down in asymmetric scenarios where a single client wishes to learn from independent providers who have no incentive to collaborate or receive updates.

Knowledge distillation (KD) provides another paradigm for model training, enabling unidirectional knowledge transfer: a student model learns from the predictions of pretrained teachers, potentially using data different from what the teachers were trained on. KD is especially attractive in settings where privacy, scalability, and minimal coordination are critical, as it allows knowledge sharing without revealing raw data or requiring ongoing interaction.

Building on this idea, we propose a one-shot federated distillation framework in which a client learns from monoclass teacher models trained independently by different providers. Each teacher, trained on private, class-specific data, is shared once and remains passive thereafter. The client aggregates these predictions with its own unlabeled data to train a multi-class student model. This approach is scalable, privacy-preserving, and coordination-free—but introduces a key challenge: out-of-distribution (OOD) supervision.

Because each teacher is trained on a single class, it often produces confident but incorrect predictions on inputs from unseen classes. Aggregating such outputs introduces noise that can impair the generalization ability of the student model.

This paper presents a systematic study of knowledge fragmentation in this one-shot distillation setting. We define five configurations with varying class coverage per teacher and empirically evaluate the impact of fragmentation on the performance of the student. Experiments on MNIST, FashionMNIST, and CIFAR-10 show that increasing fragmentation consistently degrades accuracy. To address this, we discuss three mitigation strategies: (1) exposing teachers to diverse off-class examples, (2) penalizing overconfidence in the student loss, and (3) applying contrastive learning to sharpen feature boundaries.

Our findings highlight the risks of OOD supervision in fragmented settings and propose practical strategies to improve robustness in privacy-constrained, asymmetric knowledge transfer scenarios.

The structure of the paper is as follows. The paper begins by reviewing related work on knowledge distillation, federated learning, and modular learning architectures. This is followed by a detailed presentation of the proposed client–server framework, including the training of monoclass teacher models and the distillation procedure used to construct the student model. The experimental setup is then described, with an emphasis on the design of knowledge fragmentation configurations and the evaluation protocol across multiple benchmark datasets. Subsequently, the impact of fragmentation on student model performance is analyzed, and the key limitations introduced by out-of-distribution supervision are discussed. Several mitigation strategies are then discussed. The paper concludes with a summary of findings and a discussion of directions for future work.

## 2    Related Work

This work lies at the intersection of knowledge distillation, expert specialization, and decentralized learning. Relevant literature is reviewed across three areas: (1) knowledge distillation in single and multi-teacher settings; (2) federated and decentralized learning frameworks; and (3) modular and mixture-of-experts architectures for specialization. Prior methods are analyzed in terms of communication structure, supervision format, and assumptions about label space coverage. Particular attention is given to approaches requiring full label access and the challenges introduced by fragmented class supervision. In contrast to these approaches, the method studied here operates under extreme supervision fragmentation, using one shot, unidirectional communication where only the weights of independently trained monoclass experts, each trained on a single class, are shared. The primary goal is to analyze how such fragmentation impacts the effectiveness of distillation and the resulting generalization of the student model.

## 2.1 Knowledge Distillation

Knowledge distillation (KD) trains a compact student model to mimic the predictions of a larger teacher. The foundational work by Hinton et al. (2015) introduced soft targets as supervisory signals, capturing richer information than hard labels. A broad survey of KD techniques is provided by Gou et al. (2021).

Multi-teacher KD extends this idea by aggregating predictions from multiple teachers. Fukuda et al. (2017) explored strategies like switched and augmented training to integrate teacher outputs. Zhang et al. (2022a) proposed a confidence-aware weighting scheme, assigning higher influence to more reliable teachers. These methods improve supervision but typically assume centralized training and a shared label space.

Privacy-preserving knowledge transfer methods like the Private Aggregation of Teacher Ensembles (PATE) framework (Papernot et al., 2016) mitigate data-sharing risks by training teachers on disjoint private datasets. The ensemble remains inaccessible to the student; instead, a separate student model learns from public data labeled via noisy, differentially private aggregation of teacher predictions. While PATE offers strong privacy guarantees, it assumes sufficient label coverage across teachers and requires a carefully calibrated aggregation mechanism.

These approaches demonstrate that supervision and training can be decoupled. Yet in decentralized environments with extreme supervision fragmentation, where each teacher observes only a single class, standard KD techniques become infeasible. Their reliance on shared or overlapping class coverage, bidirectional communication, or joint prediction spaces limits their applicability. The next section examines federated and decentralized paradigms that attempt to relax these assumptions.

## 2.2 Federated Learning and Decentralized Distillation

Federated learning enables distributed model training without sharing raw data. Classical methods such as McMahan et al. (2017) require synchronized communication rounds and assume shared label spaces across clients. Personalized FL methods (Fallah et al., 2020; Li et al., 2021) introduce client-specific adaptation while maintaining a global initialization, but still rely on centralized coordination and iterative updates.

To reduce communication, several FL variants adopt knowledge distillation, exchanging logits instead of model parameters. For example, FEDMD (Li & Wang, 2019) aligns local models using a shared public dataset, FEDKD (Wu et al., 2022) performs round-based logit exchanges, and FEDDF (Lin et al., 2020) ensembles local models at a central server. While reducing parameter transfer, these approaches still require auxiliary data, coordination, and label-space alignment.

One-shot and limited-round methods have emerged to further reduce overhead. FEDKT (Li et al., 2020) performs one-shot logit transfer on shared public data, FEDISCA (Kang et al., 2023) combines image synthesis with model adaptation, and DS-FL (Itahara et al., 2021) removes the server via peer-to-peer updates. However, these methods still rely on shared data, task alignment, or multiple participants interacting during training. Several recent methods explicitly address the *missing-class problem* in federated learning. FEDRS (Li & Zhan, 2021) introduces a restricted softmax mechanism that avoids updating classifier weights for unseen classes during local training, mitigating issues from label distribution skew. FEDMR (Hu et al., 2022) proposes model recombination, where layers from locally trained models are shuffled and recombined by the server to preserve local specialization and enhance global generalization. FEDLC (Zhang et al., 2022b) calibrates local logits before applying softmax, correcting for class imbalance and reducing overfitting to locally dominant classes. FEDGELA (Fan et al., 2023) addresses partially class-disjoint data by fixing a global classifier as a simplex Equiangular Tight Frame (ETF) and adapting it locally, ensuring both global consistency and local adaptability. While these methods improve robustness under heterogeneous label distributions, they rely on repeated communication, coordinated updates, or centralized aggregation. In contrast, our method performs *one-shot distillation* using independently trained monoclass experts, transmitted once to the client without coordination, label sharing, or iterative updates. This enables decentralized learning under extreme fragmentation, relaxing many assumptions made in prior federated learning work.

### 2.3 Mixture of Experts and Modular Architectures

Mixture of Experts (MoE) architectures consist of multiple specialized subnetworks, called experts, where only a subset is activated for each input (Jacobs et al., 1991). A gating mechanism dynamically selects or weights relevant experts, enabling scalable model capacity through sparse computation. Large scale implementations such as GShard (Lepikhin et al., 2020) and Switch Transformers (Fedus et al., 2022) have demonstrated strong performance on language and vision tasks by leveraging expert routing.

MoE models are typically trained end-to-end using a global loss, with both the experts and gating mechanism co-optimized over a shared dataset and unified label space. Extensions such as Modular Networks and Task Routing (Rosenbaum et al., 2019) improve modularity and interpretability, but continue to rely on centralized training, shared data, and coordinated optimization.

While the proposed method also employs specialized models, it differs fundamentally in structure and training assumptions. Experts are trained independently on private, class-specific data, without access to other classes or coordination during training. No gating mechanism or global objective is used. Instead, the softmaxed outputs of all monoclass teachers are aggregated to supervise the training of a multi-class student model via a one-shot distillation process. This approach enables modular knowledge transfer in decentralized settings, without requiring shared data or collaborative training.

### 2.4 Summary

This work explores a distillation setting where each provider trains a monoclass teacher on private data, without coordination or label sharing. These frozen models are sent to a client, which distills a global classifier in a single step, without joint training, public datasets, or bidirectional communication.

Unlike prior work in multi-teacher distillation, modular architectures, and federated learning, this setup introduces extreme label fragmentation, where supervision is narrowly scoped and decentralized. Existing methods do not address the generalization challenges introduced by such fragmentation. This gap is addressed through a systematic analysis of how label space fragmentation impacts student generalization, and by introducing a one-shot distillation strategy tailored to decentralized, privacy-sensitive environments.

## 3 Methodology

### Problem Setting

In many practical applications, valuable labeled data is distributed across multiple independent providers. However, privacy, legal, or proprietary constraints often prevent these providers from sharing raw data. While traditional approaches such as federated learning enable collaborative model training without data sharing, they often require frequent bidirectional communication and coordinated optimization, which may not be feasible in all environments.

This work considers a decentralized and asynchronous knowledge-sharing scenario. In this setting, a lightweight client aims to learn a classifier over a target set of classes by interacting with a set of providers. Each provider is willing to share trained models but not raw data. Specifically, each provider independently trains and shares a set of *monoclass teacher* models, each specialized in recognizing a single class. These teachers encapsulate local knowledge and are transferred to the client as frozen models. The client then uses the outputs of these teachers to supervise a student model through unidirectional knowledge distillation.

This setup supports privacy preservation by avoiding raw data exposure while allowing lightweight clients to benefit from distributed expertise. It also eliminates the need for coordinated model updates or gradient exchange across providers, enabling operation in low-trust, bandwidth-constrained, or asynchronous environments.

**Overview of the Approach**

The approach consists of the following components (illustrated in Figure 1): (1) a set of providers holding local datasets; (2) monoclass teachers trained to recognize individual classes while rejecting unfamiliar inputs as "other"; (3) a client-side mechanism to request and aggregate teacher outputs for a desired classification task; and (4) a student network trained to mimic the aggregated teacher signal in an unsupervised manner.

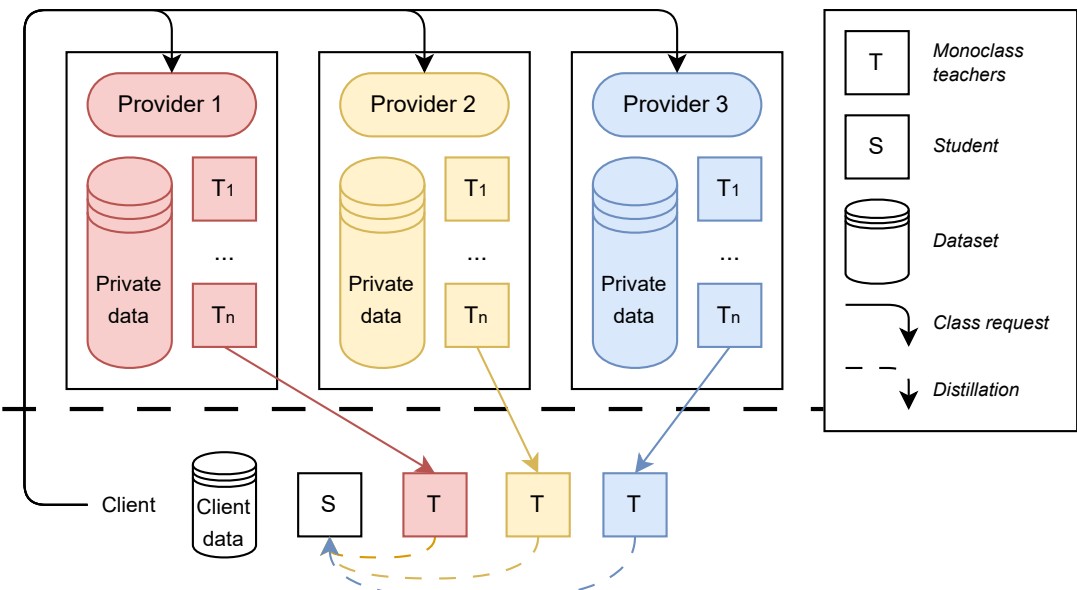

Figure 1: System overview of the proposed decentralized distillation approach. Each provider trains a set of monoclass teacher models (**T**) on private local data without sharing raw inputs. The client requests teachers for specific classes and receives frozen models. These teacher models are then used to supervise a student model (**S**) on unlabeled client data through unidirectional knowledge distillation. The legend shows the roles of each component: datasets (cylinders), monoclass teachers (**T**), student model (**S**), class request (solid arrow), and distillation flow (dashed arrow).

**Providers and Class Assignment**

Let there be $P$ providers. Each provider $p \in \{1, \ldots, P\}$ holds data corresponding to a subset of classes:

$$\mathcal{C}_p \subset \{1, 2, \ldots, K\},$$

where $K$ is the total number of classes. The class subsets $\mathcal{C}_p$ may overlap, meaning a given class $k$ can appear in the datasets of multiple providers. This redundancy can contribute to increased robustness in teacher selection and aggregation.

**Monoclass Teacher Training**

To share knowledge without exposing raw data, each provider trains a separate monoclass teacher for each supported class. Each teacher is trained to perform binary classification: distinguishing its *main* class from an aggregated *other* class composed of out-of-scope samples. Formally, for an input $x$, a teacher for class $k$ from provider $p$ outputs:

- $z_{k,\text{main}}^{(p)}(x)$: the logit corresponding to the main class.

- $z_{k,\text{other}}^{(p)}(x)$: the logit corresponding to the "other" class.

This design enables teachers to encapsulate provider-specific expertise while remaining agnostic to classes they have not seen.

**Client Request and Teacher Selection**

A client defines a set of requested classes:

$$\mathcal{C}_R \subseteq \bigcup_{p=1}^{P} \mathcal{C}_p,$$

with cardinality $|\mathcal{C}_R| = C$. For each class $k \in \mathcal{C}_R$, let

$$\mathcal{P}_k = \{\, p \mid k \in \mathcal{C}_p \,\}$$

denote the set of providers offering class $k$. When multiple providers offer the same class, the client selects a single teacher to use, following a simple criterion:

$$T(k) = \min \mathcal{P}_k,$$

i.e., the provider with the smallest index. This selection strategy can be extended in future work using confidence scores or quality metrics.

**Teacher Output Aggregation**

Each selected teacher $T(k)$ produces its main logit $z_{k,\mathrm{main}}^{(T(k))}(x)$ for input $x$. The client aggregates these logits into a unified prediction vector:

$$z_{\mathrm{agg}}(x) = \big(z_{\mathrm{agg},k_1}(x),\, z_{\mathrm{agg},k_2}(x),\, \ldots,\, z_{\mathrm{agg},k_C}(x)\big) \in \mathbb{R}^C,$$

where $z_{\mathrm{agg},k}(x) = z_{k,\mathrm{main}}^{(T(k))}(x)$ and $\mathcal{C}_R = \{k_1, k_2, \ldots, k_C\}$. For simplicity, only the main logits are used for aggregation in this work, but the "other" logits may be incorporated in future extensions via weighted fusion or uncertainty modeling.

The full aggregation and distillation process, including logit selection, softmax normalization, and student supervision, is illustrated in Figure 2.

**Student Model and Distillation Loss**

The student network receives the same input $x$ and produces a logit vector:

$$z_S(x) = \big(z_{S,k_1}(x),\, z_{S,k_2}(x),\, \ldots,\, z_{S,k_C}(x)\big) \in \mathbb{R}^C.$$

Since the client operates in an unlabeled setting, the student is trained to match the teacher-provided aggregated logits using mean squared error (MSE) loss:

$$L_{\mathrm{MSE}}(x) = \frac{1}{C} \sum_{k \in \mathcal{C}_R} \left(z_{k,\mathrm{main}}^{(T(k))}(x) - z_{S,k}(x)\right)^2,$$

or equivalently,

$$L_{\mathrm{MSE}}(x) = \|z_S(x) - z_{\mathrm{agg}}(x)\|^2.$$

This distillation approach allows the student to learn from soft signals derived from independently trained and fragmented sources without requiring access to labels or provider data.

**Reuse and Contributions.**

This approach reuses standard components of knowledge distillation, including the use of soft predictions and MSE-based student supervision. However, the key novelty lies in the one-shot federated setup using monoclass teacher models, the unlabeled aggregation of fragmented outputs, and the analysis of how such fragmentation impacts student learning. Unlike traditional distillation or federated learning setups, our method assumes no collaboration between providers and no access to ground-truth labels on the client side.

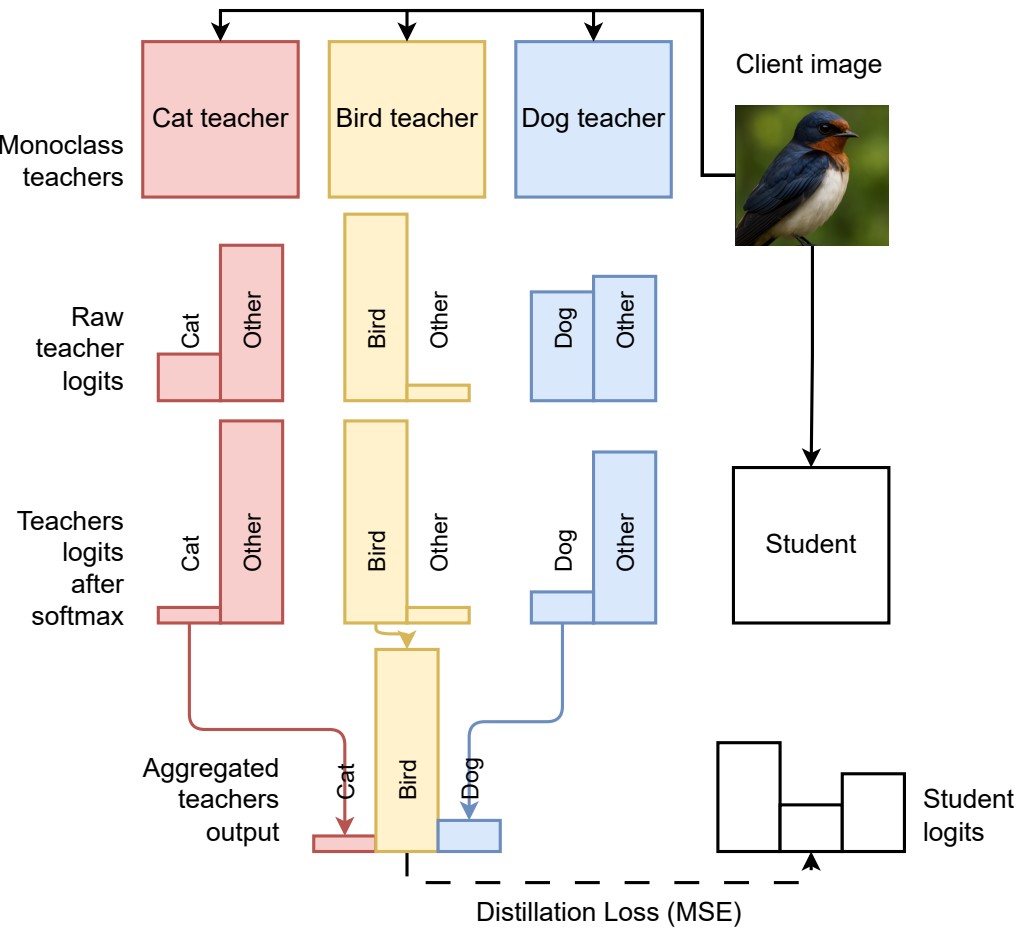

Figure 2: Illustration of the distillation process using monoclass teachers. A shared client image is passed to all relevant monoclass teachers (e.g., Cat, Bird, Dog). Each teacher produces logits for its main class and an "other" class. After applying softmax, the main logits are aggregated into a unified prediction vector. This vector supervises the student model via mean squared error (MSE) loss.

## 4 Experiments

The goal of the experiments is to evaluate the effectiveness of the proposed knowledge-sharing method in distilling accurate and generalizable models from fragmented sources of expertise. In many real-world scenarios, knowledge providers may be specialized in only a specific subset of classes. Due to privacy or resource constraints, these providers can typically share only teacher models trained on their local, limited data.

The proposed method addresses this challenge by aggregating outputs from specialized teachers. These teachers may come from different providers, each exposed to a different subset of classes. Their predictions are used to train a single unified student model. A central difficulty lies in integrating this fragmented knowledge. Predictions may be inconsistent, uncalibrated, or biased due to the varying and limited training scopes of individual providers.

### 4.1 Motivation

This work focuses on scenarios with high knowledge fragmentation, where each provider contributes only one or a few monoclass teachers. Such settings reflect practical constraints in domains like healthcare, industry, or edge-device learning. In these contexts, privacy, bandwidth, or computational limitations often prevent centralized training.

The proposed approach is intended for deployment. Both teacher and student models are compact and suitable for resource-constrained environments, such as mobile devices or IoT platforms. Demonstrating effective generalization under these constraints is a central objective.

### 4.2 Experimental Objectives

The experiments are designed to answer the following questions:

- **How well can a student learn from fragmented supervision?**

- **How does fragmentation intensity affect generalization?**

- **Do stronger monoclass teachers improve aggregation quality?**

- **How close can the student get to fully supervised performance?**

### 4.3 Experimental Setup

Evaluation is conducted on three standard image classification benchmarks: MNIST, FashionMNIST, and CIFAR-10, representing increasing levels of difficulty. Each dataset is split into training and test sets. From the training set, 80% is used to train teacher models, and 20% is reserved as unlabeled data for the client (student). Only the outputs from the teachers are available to the student during training.

**Model Architecture.** Both teacher and student models use the same compact convolutional architecture. Designed for deployment in low-resource environments, it comprises fewer than 100K parameters while maintaining strong representational capacity. The architecture features two convolutional blocks with batch normalization and pooling, followed by a lightweight fully connected head. This shared design ensures consistency in capacity and learning dynamics across both roles. The complete layer configuration is summarized in Table 1.

**Teacher and Student Setup.** Each teacher is trained as a binary classifier to distinguish one target class from all others. Its two output logits correspond to the assigned class ("main", label 0) and the rest ("other", label 1). The student is a standard multiclass classifier trained to predict across all classes.

| Layer | Type | Output Shape | Details |
|---|---|---|---|
| Input | - | (1 or 3, 32, 32) | Depends on dataset |
| Conv1 | Conv2D | (8, 32, 32) | 8 filters, kernel size 3, padding 1 |
| BN1 | BatchNorm2D | (8, 32, 32) | - |
| ReLU + MaxPool | Activation + Pool | (8, 16, 16) | Pool size 2x2 |
| Dropout1 | Dropout | (8, 16, 16) | $p = 0.3$ |
| Conv2 | Conv2D | (16, 16, 16) | 16 filters, kernel size 3, padding 1 |
| BN2 | BatchNorm2D | (16, 16, 16) | - |
| ReLU + MaxPool | Activation + Pool | (16, 8, 8) | Pool size 2x2 |
| Dropout2 | Dropout | (16, 8, 8) | $p = 0.3$ |
| Flatten | - | (1024) | Flatten from $16 \times 8 \times 8$ |
| FC1 | Linear | (16) | Followed by LayerNorm and ReLU |
| Dropout3 | Dropout | (16) | $p = 0.3$ |
| FC2 | Linear | (10) | Number of output classes |

Table 1: Architecture of the model used for both teachers and student.

**Teacher Output Aggregation.** At inference, each teacher's logits are passed through a softmax over its two outputs. We extract the softmax probability corresponding to the "main" class and concatenate these across all teachers. This produces a pseudo-logit vector of dimension $N$ (number of teachers), used as a soft target to train the student using mean squared error (MSE) loss.

**Training Protocol.** Teachers are trained for 200 epochs with cross-entropy loss, without class weighting. The student is trained for 200 epochs with MSE loss on aggregated soft outputs. Both use the Adam optimizer with a learning rate of 0.001. Batch size is 2048 for teachers and 512 for the student.

**Supervised Client Baseline.** As a reference, the same architecture is trained using labels from the client split. This provides a supervised baseline for comparison with the distillation-based student.

### 4.4 Fragmentation Configurations

Five fragmentation configurations are defined to simulate varying degrees of knowledge distribution across providers. Each provider (i.e., teacher) is trained on a specific subset of classes. As fragmentation increases, the number of providers grows and each teacher's view of the full class distribution becomes more limited.

| Setup | Providers | Classes/Provider | Class Assignments |
|---|---|---|---|
| A | 1 | 10 | [0,1,2,3,4,5,6,7,8,9] |
| B | 2 | 5 | [0,1,2,3,4], [5,6,7,8,9] |
| C | 4 | 4 | [0,1,2,3], [3,4,5,6], [6,7,8,9], [9,0,1,2] |
| D | 4 | 3 | [0,1,2], [3,4,5], [6,7,1], [8,9,2] |
| E | 5 | 2 | [0,1], [2,3], [4,5], [6,7], [8,9] |

Table 2: Fragmentation setups showing the number of providers, classes per provider, and class assignments.

Setup A represents centralized training with full class access. Setup E corresponds to the maximum fragmentation, where each provider sees only two disjoint classes. Intermediate setups simulate increasing specialization and overlap.

### 4.5 Evaluation Metrics

The method is evaluated using the following metrics:

- **Binary Teacher Accuracy:** Each teacher is trained to detect one specific class. It performs a binary classification task. The output is 0 for the main class and 1 for the "other" classes. The reported accuracy is the average of the accuracy on the main class and the "other" class.

- **Aggregation Confusion Matrix:** Predictions are obtained from the concatenated "main class" probabilities across all teachers. A confusion matrix is computed to assess the quality of the aggregated ensemble output prior to student training.

- **Student Accuracy:** The student top-1 accuracy on the test set measures how well it generalizes from teacher outputs without label supervision.

- **Supervised Client Accuracy:** Accuracy of the same model trained with labeled client data. This serves as a performance upper bound under the same data budget.

### 4.6 Summary

These experiments quantify the trade-offs between model specialization and generalization under fragmented supervision. By distilling knowledge from multiple narrow teacher models into a unified student, the feasibility of label-free generalization in deployment-constrained settings is demonstrated. The experimental setup isolates the effects of fragmentation and enables evaluation of how aggregation quality impacts student performance.

## 5 Analysis of Knowledge Fragmentation Impact

The effect of knowledge fragmentation is evaluated by measuring teacher accuracy, aggregation quality, and student generalization across increasing levels of fragmentation. Five fragmentation setups are tested on MNIST, FashionMNIST, and CIFAR-10. These datasets differ in complexity and class similarity, providing a broader view of how fragmentation impacts performance.

### 5.1 MNIST: Low Complexity Domain

In MNIST (Figure 3), teacher models retain high accuracy across all fragmentation levels. Performance decreases slightly, from 99.27% in the full-data setup to 79.88% in the most fragmented configuration. This shows that local learning remains strong even with limited class coverage. In contrast, aggregation accuracy drops more sharply, from 99.12% to 42.39%, indicating that outputs from disjoint teachers become harder to combine as class overlap decreases.

Student performance closely follows the decline in aggregation quality. Accuracy drops from 98.46%, near the full-data baseline of 98.36%, to 44.47%. The student model fails to generalize when trained on inconsistent and fragmented aggregated outputs. These results suggest that, in low-complexity settings, fragmentation primarily degrades the aggregation step.

Figure 4 shows progressive degradation in prediction coherence. Setup A is nearly diagonal, indicating strong alignment. Setup B introduces slight confusion (e.g., 3 vs. 5), while setup C shows increased confusion for shared digits like 3, 7, and 9. In setups D and E, matrices become noisier—digit 1 remains clear, but digit 2 is misclassified in setup D despite multiple appearances. By setup E, the class structure collapses, and aggregation fails to unify teacher outputs.

### 5.2 FashionMNIST: Medium Complexity Domain

In FashionMNIST (Figure 5), teacher accuracy remains relatively high, decreasing from 95.43% to 74.34% across setups. Aggregation accuracy, however, shows a more severe drop—from 92.86% in the full setup to

44.88% in the most fragmented case. This decline reflects the difficulty of merging outputs when classes are visually similar and rely on shared context for disambiguation.

Student accuracy is affected in the same way. Performance decreases from 89.14% to 44.47%. In earlier setups, the student performs near the baseline of 89.31%, but generalization fails when aggregation becomes unreliable. These results suggest that in medium-complexity tasks, fragmentation amplifies inconsistency among teachers, especially for fine-grained class distinctions.

In Figure 6, setup A shows strong class separability, with predictions aligned along the diagonal. Setup B introduces confusion among visually similar classes like shirts (6), coats (4), pullovers (2), and T-shirts (0). In setup C, this confusion intensifies, particularly for coats and T-shirts, due to inconsistent predictions from overlapping groups. Setups D and E show widespread misclassifications, with shirts and pullovers frequently confused, and structure breaking down entirely in the disjoint setting.

### 5.3   CIFAR-10: High Complexity Domain

In CIFAR-10 (Figure 7), teacher accuracy starts at 80.72% and falls to 66.22% as fragmentation increases. Aggregation suffers a stronger decline, dropping from 74.28% to 29.46%. Visually similar classes (e.g., cats vs. dogs, trucks vs. cars) contribute to high ambiguity across teachers. When context is lost, aggregation consistency is severely reduced.

Student accuracy starts at 61.18%, near the baseline (61.08%), but drops sharply to 29.94%. Fragmentation disrupts not only the integration of local predictions but also the formation of global class boundaries by the student. Aggregated outputs lose coherence, and learning becomes unstable.

In Figure 8, setup A shows relatively good class separability. In setup B, misclassifications increase sharply across the boundary, particularly between classes 3-5, 0 and 8 or 1 and 9. This indicates aggregation challenges across disjoint providers. In setup C confusion persists among shared animal classes (e.g., 3, 5, 6), suggesting misaligned knowledge despite partial redundancy. In setup D, narrow overlaps lead to severe ambiguity for classes like cats (3), deer (4), and birds (2), which are spread across multiple groups. Finally, setup E, with disjoint class pairs, exhibits the most severe degradation.

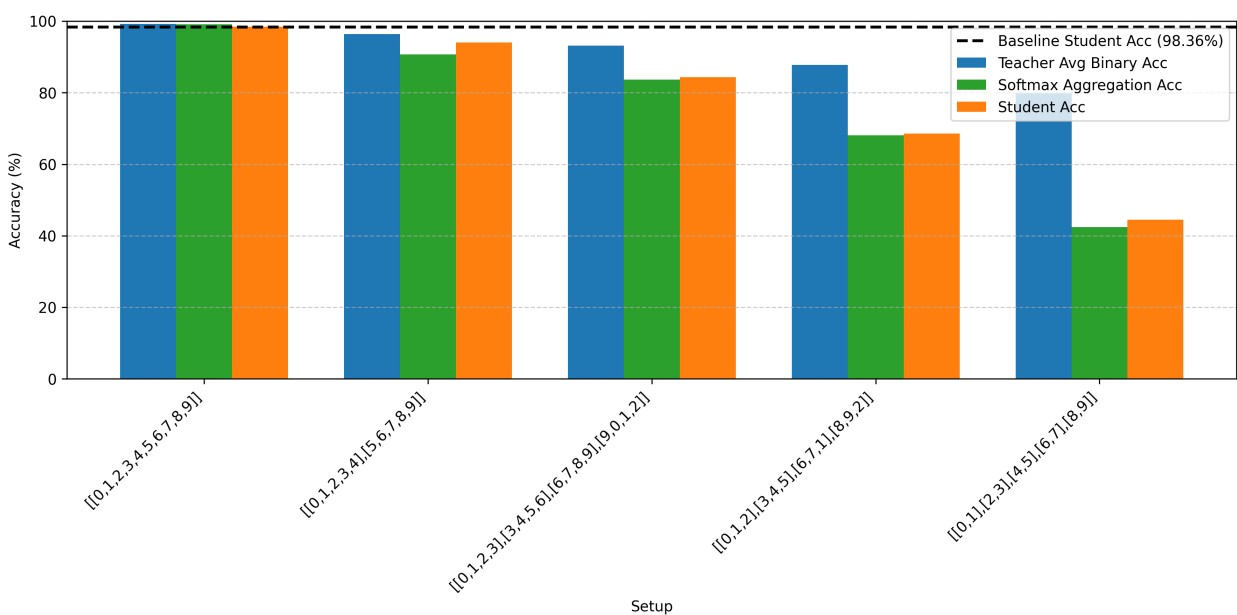

Figure 3: Comparison of teacher, aggregation, and student accuracy across fragmentation setups for MNIST.

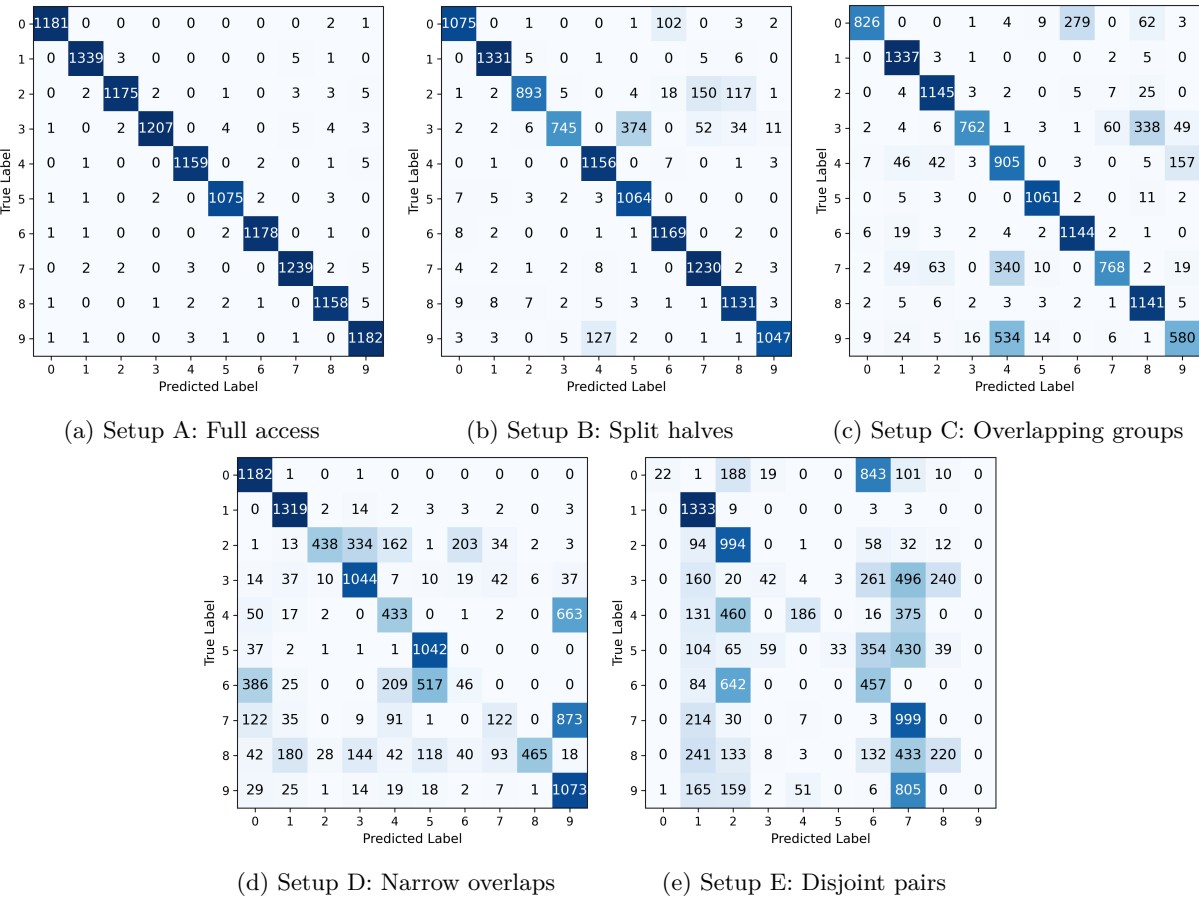

Figure 4: Confusion matrices of aggregated teacher outputs for MNIST across fragmentation setups.

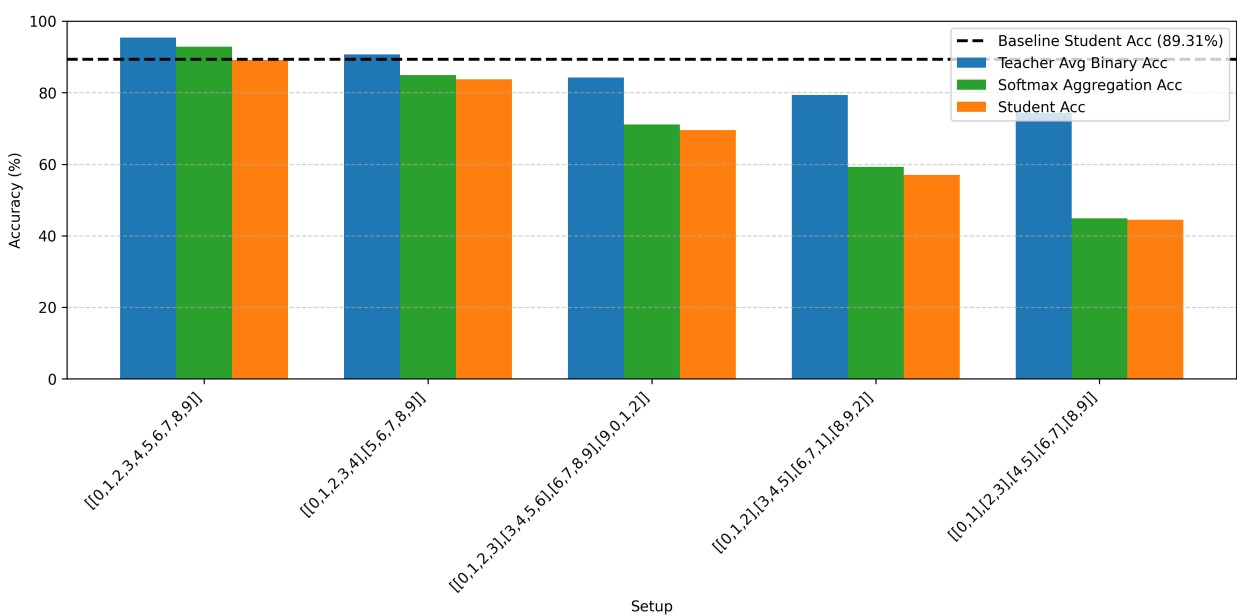

Figure 5: Comparison of teacher, aggregation, and student accuracy across fragmentation setups for FashionMNIST.

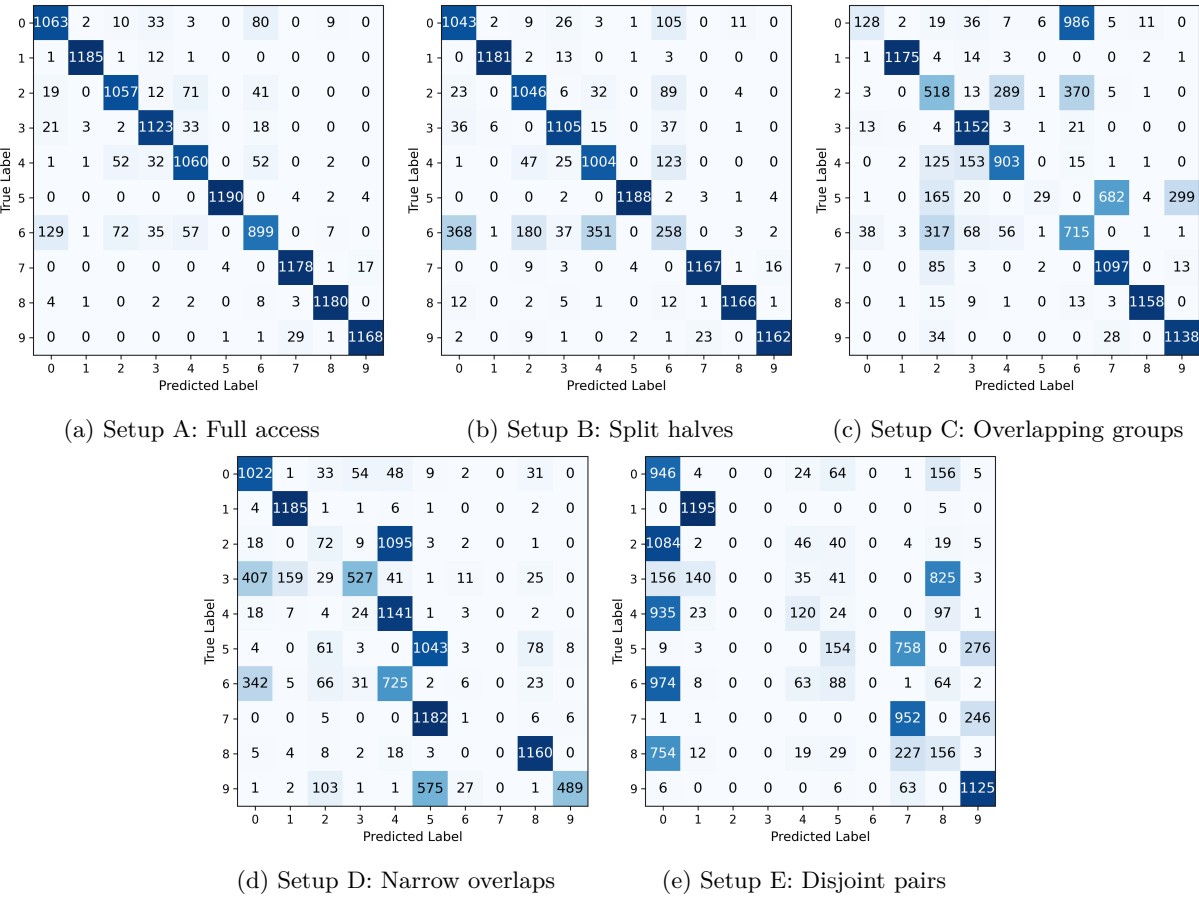

Figure 6: Confusion matrices of aggregated teacher outputs for FashionMNIST across fragmentation setups.

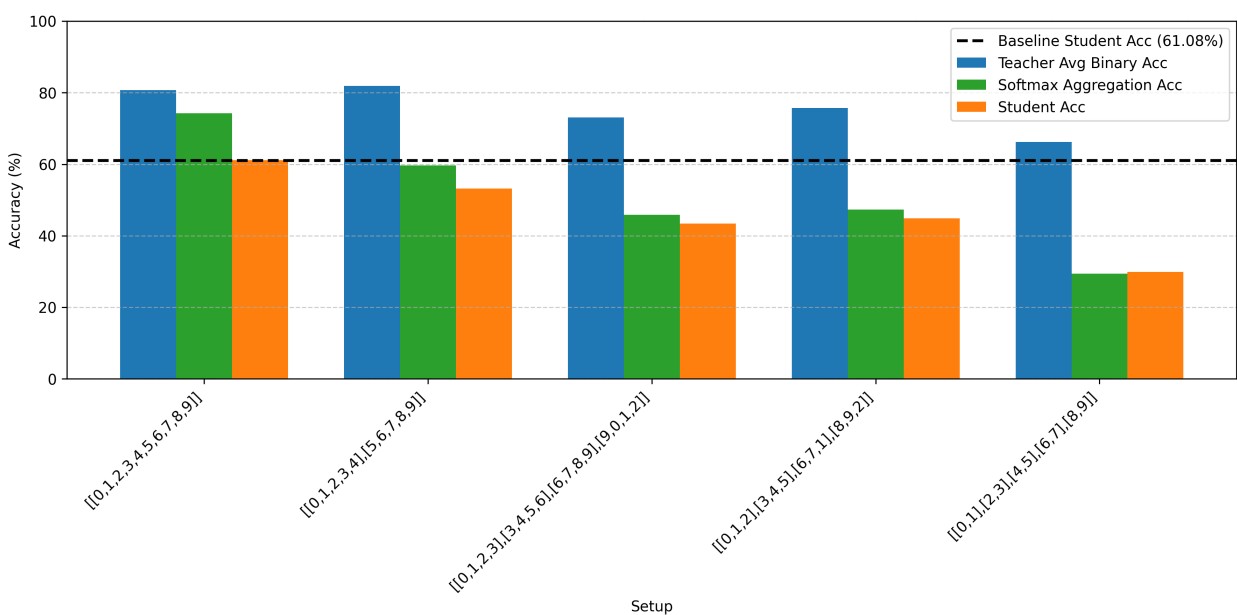

Figure 7: Comparison of teacher, aggregation, and student accuracy across fragmentation setups for CIFAR-10.

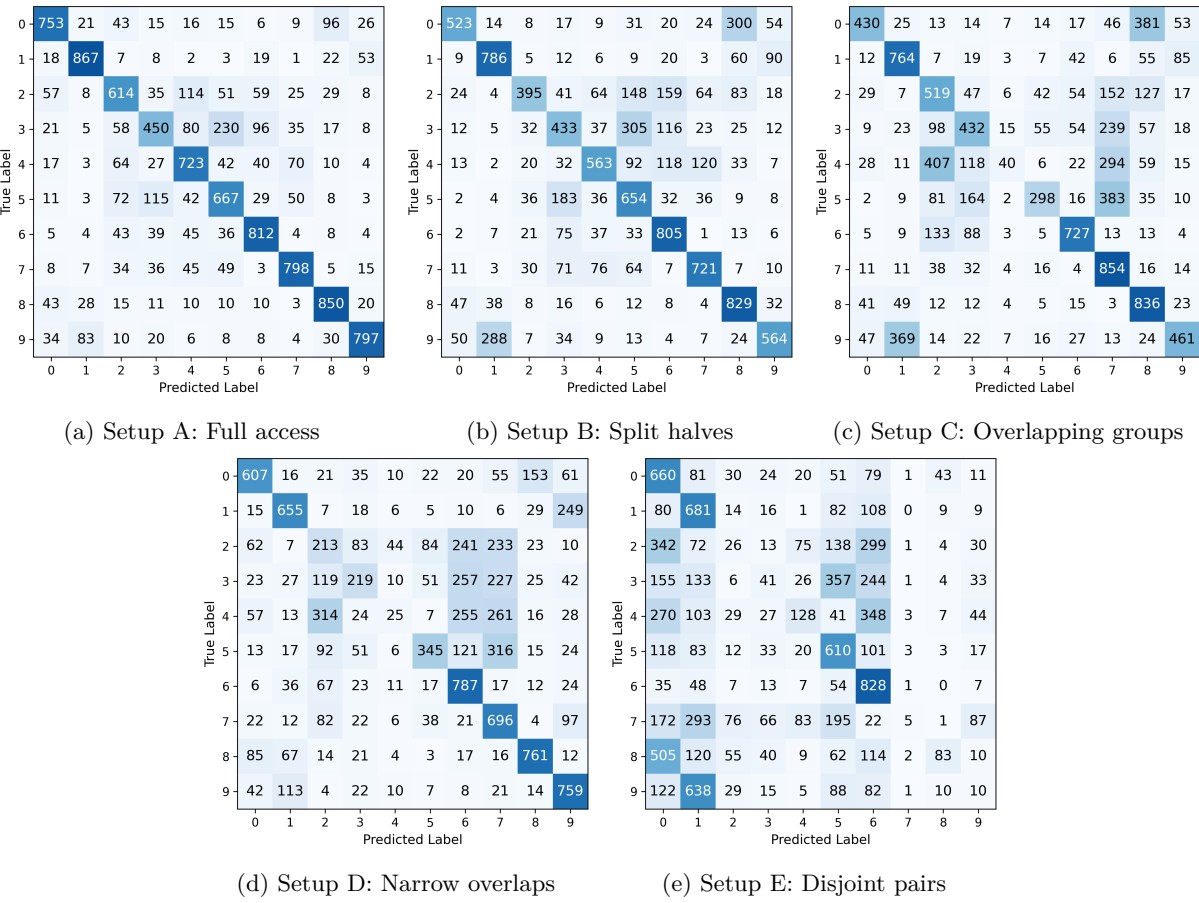

Figure 8: Confusion matrices of aggregated teacher outputs for CIFAR-10 across fragmentation setups.

### 5.4 Discussion

The results highlight a consistent and significant impact of knowledge fragmentation across all datasets. As fragmentation increases, student performance declines sharply, despite relatively stable teacher accuracy. This drop is primarily driven by degraded aggregation quality.

In MNIST, a low-complexity dataset with clearly separable classes, teacher accuracy remains high across all setups. Aggregation performance declines more rapidly, especially with disjoint or narrowly overlapping subsets. Still, the task structure allows some generalization, and student accuracy stays moderate until extreme fragmentation.

FashionMNIST shows more subtle inter-class differences. Teachers remain locally accurate, but aggregation becomes less reliable as fragmentation increases. Visual similarity between classes such as shirts, coats, and pullovers leads to confusion when global context is lost. Student performance drops earlier and more sharply than in MNIST.

CIFAR-10 shows the strongest sensitivity. Even with full data, aggregation and student accuracy are lower due to visual complexity and semantic overlap between classes. Fragmentation quickly leads to inconsistent aggregated outputs, and student learning collapses under moderate-to-high fragmentation. The teacher ensemble fails to provide coherent supervision.

Across datasets, fragmentation produces noisy aggregated outputs. This is not caused by poor teacher accuracy on their assigned classes, but by their inability to correctly handle out-of-scope inputs. The bottleneck lies in the quality of the outputs from the teachers, which directly impact aggregation. As a result, the student receives inconsistent supervision and fails to generalize. Mitigating this requires reducing noise at the teacher level, particularly in how unseen classes are treated.

While privacy motivates our decentralized and communication-free design, it is not empirically evaluated in this work. Our approach assumes white-box access to teacher models and does not incorporate formal privacy-preserving mechanisms. Future work could explore integrating differential privacy (DP), secure enclaves, or cryptographic aggregation to strengthen protection against privacy leakage.

As the number of classes increases, we expect fragmentation effects to intensify. Larger label spaces increase the likelihood of both semantic overlap between classes and severe supervision gaps across experts. In such settings, monoclass teachers are more likely to produce conflicting or overconfident predictions on out-of-scope examples, further degrading aggregation. While our current experiments are limited to ten-class datasets, extending this approach to more fine-grained or large-scale classification tasks (e.g., CIFAR-100, ImageNet) remains an important direction for future work. Mitigation strategies such as hierarchical class grouping, broader "other" class exposure, or adaptive aggregation will likely become even more critical in such settings.

## 6 Improving Teacher Generalization under Fragmented Supervision

In highly fragmented settings, each teacher is exposed to a limited subset of the label space. Without explicit mechanisms to distinguish between in-class and out-of-class inputs, teachers often produce overconfident or inconsistent outputs on unfamiliar samples. These errors compound during aggregation, reducing supervision quality and harming generalization.

To address this, several strategies can be adopted to improve teacher robustness and reduce noise in their outputs, particularly when encountering out-of-scope inputs.

**1. Expose Teachers to a Rich and Varied "Other" Distribution**  To improve the ability of teachers to reject inputs outside their assigned classes, training should include a diverse set of examples grouped into a generic 'other' class label. These samples should span a wide range of shapes, textures, and semantics, encouraging teachers to build robust and discriminative boundaries. Unlike random negatives, this curated "other" distribution increases coverage of the input space and helps reduce overconfident misclassification on unfamiliar samples. Such data can be sourced from unassigned training classes or external datasets.

**2. Shape the Loss to Improve Robustness on "Other" Samples**  To increase robustness, the loss function can be adapted to handle the diverse and ambiguous nature of "other" samples more effectively. Beyond standard classification loss, additional terms can be used to encourage uncertainty, penalize overconfidence, or emphasize margin separation between assigned and non-assigned classes. This tailored objective helps the teacher produce more stable outputs when facing unfamiliar data, reducing misclassification and improving the quality of aggregation.

**3. Apply Contrastive Learning to Enhance Representations**  Contrastive learning can be used within each teacher to improve feature separation between assigned classes and "other" samples. By pulling together representations of in-class examples and pushing away those from the "other" category, the teacher learns more discriminative and robust embeddings. This leads to better class boundaries and reduces confusion when facing unfamiliar inputs, ultimately supporting more consistent aggregation.

These strategies aim to mitigate the core challenges of fragmented supervision by making teachers more aware of what lies outside their expertise. In doing so, they produce softer, more calibrated outputs on unfamiliar inputs—ultimately enabling the student to learn from a more stable and informative aggregated signal.

## 7 Conclusion

We studied the impact of knowledge fragmentation on one-shot federated distillation using monoclass teachers. Through systematic experiments on MNIST, FashionMNIST, and CIFAR-10, we showed that increasing class fragmentation significantly degrades student performance, with effects intensifying as visual complexity grows.

Our analysis revealed that fragmented supervision introduces ambiguity and inconsistency in the aggregated predictions. Although individual teachers perform well on their assigned classes, their lack of exposure to diverse inputs leads to overconfident and unreliable outputs on unfamiliar examples, ultimately impairing the distillation process.

To mitigate these effects, we discussed several strategies, including exposing teachers to a broader "other" class during training, penalizing overconfidence, and enhancing feature separation through contrastive learning. These approaches aim to enhance the robustness of teachers and the stability of aggregation under fragmented supervision.

Overall, our findings highlight critical limitations in naively combining narrow experts for knowledge distillation. In future work, we plan to further refine mitigation techniques and investigate more adaptive aggregation methods to support knowledge transfer in highly fragmented, privacy-sensitive environments.

## Broader Impact

This work proposes a privacy-preserving one-shot distillation method that enables decentralized model training using monoclass teacher models. While the method avoids raw data sharing and reduces coordination overhead, it may lead to overconfident predictions on out-of-distribution inputs when teachers are trained on narrow class subsets. These risks are particularly relevant in sensitive applications such as healthcare or surveillance. Future applications should incorporate uncertainty calibration or robust fallback strategies to prevent misuse.

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
