# OpenReview forum: "One-Shot Federated Distillation Using Monoclass Teachers: A Study of Knowledge Fragmentation and Out-of-Distribution Supervision"
_TMLR — Accepted by TMLR_

### Review · Reviewer_9Jpp · 2025-05-20

**Summary Of Contributions:**

This work proposes a one-shot federated distillation method where a client learns a multi-class student model by aggregating knowledge from monoclass teacher models provided by multiple data sources. A key challenge is out-of-distribution (OOD) supervision, where monoclass teachers mispredict unseen inputs, leading to noisy signals that degrade student performance. Experiments on MNIST, FashionMNIST, and CIFAR-10 show that increased knowledge fragmentation reduces student accuracy. To mitigate this, the authors propose three strategies: exposing teachers to diverse off-class examples, penalizing overconfidence, and using contrastive learning to sharpen feature boundaries.

**Audience:**

Yes

**Claims And Evidence:**

Yes

**Requested Changes:**

1) The manuscript needs to include appropriate citations for references throughout, particularly in the introduction and related work sections.
2) The effectiveness of the proposed method should be verified with scenarios involving more classes.
3) As for other minor requested changes, please see weaknesses part.

**Strengths And Weaknesses:**

Strengths:
1) Well motivated. The missing class problem among federated clients is a common and critical challenge in the development of federated learning algorithms. The author tries to solve this problem through one-shot distillation.
2) Clear presentation. The contents in the paper are clear and easy to follow.
3) Performance shown in Figure5, and Figure 7 seems good.

Weaknesses:
1) The paper lacks sufficient reference citations. For example, the entire introduction section has no references to support their claims. Additionally, the missing class problem is a widely studied topic, but the paper does not cite relevant works in this area. To name a few:
* [ICML]FedAWS: Federated learning with only positive labels
* [KDD]FedRS: Fedrs: Federated learning with restricted softmax for label distribution non-iid data
* [TMLR]FedMR: Federated learning under partially class-disjoint data via manifold reshaping
* [ICML]FedLC: Federated learning with label distribution skew via logits calibration
* [NeurIPS]FedGELA: Federated learning with bilateral curation for partially class-disjoint data

2) I am concerned about the applicability of this method to scenarios with a larger number of classes. It would be helpful to evaluate the method on datasets like TinyImageNet, which contains 100 or more classes, to assess its scalability and performance.
3) The mention of "modular learning architectures" in the last paragraph of the introduction seems unclear. It is not explained why this approach is introduced or how it helps address the problem. Providing a clearer rationale and connection to the problem would improve understanding.
4) There appears to be an issue with the manuscript's flow between Sections 5.3 and 5.4.

---

### Review · Reviewer_fPw7 · 2025-05-27

**Summary Of Contributions:**

This paper focuses on the challenge of knowledge transfer in distributed machine learning environments where data sharing is constrained by privacy, legal, or proprietary concerns.  The authors consider the knowledge distillation (KD), which allows a student model to learn from
teacher predictions.

Specifically, the authors propose a one-shot federated distillation framework using monoclass teacher models. In this setup, each provider independently trains a set of teacher models, each specialized in a single class, and shares the frozen model weights once with a client. The client then distills a global multi-class student using only the predictions of these monoclass teachers on unlabeled local data.

Experiments on MNIST, FashionMNIST, and CIFAR-10 show that increasing fragmentation consistently degrades accuracy. To address this, the authors explore three mitigation strategies: (1) exposing teachers to diverse off-class examples, (2) penalizing overconfidence in the student loss, and (3) applying contrastive learning to sharpen feature boundaries.

**Audience:**

Yes

**Broader Impact Concerns:**

The research motivation is particularly relevant in sensitive applications such as healthcare or surveillance.

**Claims And Evidence:**

No

**Requested Changes:**

- Clearly state what is new compared with prior works in federated and knowledge distillation (particularly in relation to the "one-shot" and "fragmented" settings). Outline concrete distinctions from survey-cited methods.
- Include experimental results and quantitative comparisons with established baselines (e.g., classical federated distillation, multi-teacher KD, or recent privacy-preserving distillation protocols such as those reviewed in [1]).
- Explicit Attribution: For each methodological component, specify if it is adopted from existing literature, or describe succinctly how it departs from precedent.
- To support the privacy motivation, add empirical or theoretical analyses such as privacy risk/attack experiments, ablation of privacy mechanisms, or mention of compatibilities with formal frameworks (e.g., differential privacy).
- Elaborate on the discussion in Section 6 with concrete evidence.

**Strengths And Weaknesses:**

### Strengths

+ This paper explores an interesting topic on a one-shot federated distillation method. The paper addresses the important and timely problem of privacy-preserving knowledge transfer and model distillation in federated and decentralized environments.
+ The authors show some experimental studies on the proposed method across multiple datasets.


### Weakness
- The primary concern is that the core innovative contribution is neither clearly stated nor sufficiently differentiated from prior literature. The related work discussion is general. It remains unclear which particular direction the authors follow. Moreover, this paper lacks comparative baselines. Thus, it is unclear whether the proposed method has advantages compared with the existing methods.
- Although the methodology in section 3 explained each step of the proposed method. It is unclear which part is new and which part is from the current literature.
- Comparative experiments against existing baselines are lacking. It is important to compare the proposed method with state-of-the-art methods, such as  [1].
- Additionally, the proposed idea is motivated by improving privacy. However, the experimental part lacks the attacking tests or privacy verification. That is, the claimed privacy properties are not empirically supported or discussed in terms of formal privacy guarantees (e.g., differential privacy, membership inference resistance).

[1] Federated Distillation: A Survey. Lin Li, Jianping Gou, Baosheng Yu, Lan Du, Zhang Yiand Dacheng Tao.

- The discussions on "improving teacher generalization" (Section 6) are general. More experimental results to support the discussion are expected.

---

### Review · Reviewer_oT3w · 2025-06-22

**Summary Of Contributions:**

This draft has confusing motivation. For example, it states, “However, privacy, legal, and proprietary concerns often limit direct data sharing.” Additionally, “While federated learning (FL) enables distributed model training, it typically assumes mutual benefit, requires repeated communication, and produces a shared global model.” It is strange that FL is introduced with "However" since it also has privacy and sharing data issues.  Then, it suddenly shifts to “one-shot federated distillation.” Later, the focus changes to the OOD setting. I cannot continue my review.

The authors are required to state their question clearly before the review process.
The quality of this draft clearly lowers the standards of TMLR.

**Audience:**

No

**Claims And Evidence:**

No

**Requested Changes:**

It would benefit from a more coherent organization and a deeper exploration of the idea.

**Strengths And Weaknesses:**

The structure is unclear, and the idea of simple teacher selection is underdeveloped.

---

> ### Comment · Action_Editor_7e1H · 2025-07-28
>
> Hi reviewer oT3w,
>
> Can you check the rebuttal and see if the new structure of their introduction looks clearer to you now? Please also submit your official recommendation as soon as possible (1 week overdue). Thank you!
>
> AE

---

### Author Response · Authors · 2025-06-23
**Author Response to Reviews for Submission #4845**

## Response to Reviewers

We thank the reviewers for their thoughtful and constructive feedback. Due to the end of the first author’s PhD contract, we are unable to run new experiments. However, we have revised the manuscript to clarify motivation, state contributions, expand citations, and improve structure and positioning.

---

### Reviewer oT3w

> **Comment:** The motivation is confusing; the transition from FL to one-shot distillation and then to OOD is unclear.

**Response:**
We agree the original introduction was unclear. We intended to motivate our setting as an alternative to collaborative FL when repeated communication is infeasible. We now structure the introduction as follows:

1. FL enables privacy-preserving training, but assumes repeated collaboration.
2. KD allows one-way knowledge transfer, enabling looser coupling.
3. Our setting involves non-collaborative providers sharing monoclass teacher models once.
4. This creates OOD supervision issues for the student.
5. Our study investigates how fragmentation affects this setup and how to mitigate it.

**We also revised phrasing throughout the section to clarify the shift from privacy concerns to the core technical focus: learning under fragmented, asymmetric supervision.**

---

### Reviewer fPw7

> **Comment:** The core innovation is unclear; related work is too general.

**Response:**
We now clearly state that our novelty lies in:
- A one-shot federated distillation setting with monoclass teachers;
- A study of knowledge fragmentation and OOD effects;
- A discussion of three conceptual mitigation strategies.

We also expanded the related work section to distinguish our approach from classical FL, multi-teacher KD, and privacy-preserving distillation.

> **Comment:** What is new vs. reused in the methodology?

**Response:**
We now clearly annotate in Section 3 that standard KD loss and soft predictions are reused, while the one-shot monoclass setup and unlabeled student aggregation are our core contributions.

> **Comment:** Lack of comparative baselines.

**Response:**
We acknowledge this limitation. Due to time constraints before the thesis defense, additional experiments were not feasible. We have added conceptual comparisons to relevant baselines in the related work section. **We also highlight how our assumptions (one-shot, fragmented, monoclass) differ fundamentally from many existing frameworks.**

> **Comment:** No empirical privacy evaluation.

**Response:**
We clarify that privacy is a motivating constraint, but not the focus. The paper studies OOD effects of fragmented knowledge. We acknowledge white-box exposure and outline future directions (e.g., DP training, secure enclaves) in the Discussion. **We also explicitly caution that our setup does not currently prevent privacy attacks, and suggest integrating formal protections in follow-up work.**

> **Comment:** Section 6 is too general; more evidence expected.

**Response:**
We cannot provide new experiments due to time constraints. However, we now frame Section 6 as forward-looking proposals and cite supporting literature showing that exposure to auxiliary samples, loss shaping, and contrastive learning can help generalization under fragmentation. **These techniques are not part of our core experiments but reflect plausible extensions that future work could evaluate.**

---

### Reviewer 9Jpp

> **Comment:** Missing references on the missing-class problem.

**Response:**
We have updated the introduction and related work to include FedAWS, FedRS, FedMR, FedLC, and FedGELA, and positioned our work as a one-shot, non-collaborative alternative.

> **Comment:** Unclear scalability to datasets with more classes.

**Response:**
We discuss this explicitly in the results section. Fragmentation severity likely increases with class count, reinforcing the need for mitigation—an avenue for future work. **We also comment on the risk of greater semantic overlap in large-class settings, which may further degrade aggregation.**

> **Comment:** “Modular learning architecture” is vague.

**Response:**
We now clarify it refers to the reusability of monoclass teachers—each trained independently and shareable without synchronization—supporting decentralized deployment.

> **Comment:** Flow issue between Sections 5.3 and 5.4.

**Response:**
We revised the end of Section 5.3 to summarize key takeaways and motivate Section 5.4 as a shift from results to analysis and mitigation.

---

### Concluding Remarks

We thank the reviewers again. While limited in resources, we believe the revised manuscript better articulates our contributions, improves clarity, and addresses the main concerns.

---

> ### Comment · Action_Editor_7e1H · 2025-07-30
>
> Hi authors,
>
> I think you had a white space in "Out-of- Distribution" in the title of this submission and I believe this white space is a typo. Please fix it later.
>
> AE

---

> > ### Comment · Action_Editor_7e1H · 2025-07-30
> >
> > I meant the title in the meta info in our openreview system, not the submitted pdf file.

---

### Decision · Action_Editor_7e1H · 2025-07-30

**Recommendation:** Accept as is

**Additional Comments:**

After downweighing some uninformative review, I think the other two reviewers were generally positive, and this paper looks interesting to some deep learning researchers/practitioners.

**Audience:**

Yes

**Audience Explanation:**

The studied problem is increasingly important in practice.

**Claims And Evidence:**

Yes

**Claims Explanation:**

There are extensive experiments, some added during rebuttal.